# Less Water, Less Oil: Policy Response for the Kenyan Future, a CGE Analysis

Davide Bazzana [1,2,*], Aidin Mobasser [1,2] and Sergio Vergalli [1,2]

1 Department of Economics and Management, University of Brescia, 25122 Brescia, Italy
2 Fondazione Eni Enrico Mattei, 20123 Milan, Italy
* Correspondence: davide.bazzana@unibs.it

**Abstract:** The continuous depletion of nonrenewable natural resources and climate change may lead to a future characterized by a higher frequency of extreme natural events (i.e., flooding, hurricanes, and droughts) and resource supply shocks (i.e., oil price shock). Sub-Saharan African countries will be particularly exposed to these types of shock due to their socioeconomic conditions and geographical conformation. This study investigates the impact of two contemporaneous covariant sudden shocks (i.e., drought and price oil shock) and the possible coping strategies through a static computable general equilibrium (CGE) model for Kenya. The results suggest that a mitigation policy as public transfers is an effective mitigation tool for drought effects, improving welfare and GDP in the short run. However, adopting public transfers during an oil crisis may have regressive effects on population income and welfare. Because the mitigation effectiveness is strongly affected by the complex interaction of combined shocks, the public authorities should pay attention to policy implementation. These findings call for a new scheme of transfer allocation where rural and low-income household quantiles should receive more attention by postdrought mitigation policy, being that they are more vulnerable to external shocks.

**Keywords:** computable general equilibrium; climate change; drought policy; oil policy; Kenya





## 1. Introduction

Developing countries, particularly sub-Saharan African countries, have been exposed to various risks and economic shocks over the past century [1]. These events can be wars, financial crises, or extreme weather conditions due to climate change [2]. When the shocks impact the broader population scale (from significant communities of people to the whole country's population) or register a wider socioeconomic impact, these shocks can be defined as covariate shocks [3]. Covariate risks concern communities, whereas idiosyncratic risks affect one or few households [2]. Idiosyncratic risks are more pronounced among the poor, rural areas due to their dependencies to the agriculture product and their sensitivity to the food price inflation alongside their higher populated household units [4]. Coping strategies for sudden covariate shocks in poor or semipoor African countries have proven to be vital in terms of welfare because of the negative impact of the shock on welfare and the economy. Since most of the population in sub-Saharan African countries lives in rural areas with low-income families, any shock could have a disastrous impact on welfare. For instance, climate change shocks, such as droughts or floods, negatively affect agricultural productivity. Reduction in agricultural productivity, particularly rain-fed production, negatively impacts most African countries that depend heavily on these agricultural products. Moreover, economic shocks (e.g., strong price oscillations) can reduce the ability of rural communities to assess food security and sustainable livelihood by self-consumption or accessing the market commodities [5].

Although the Kenyan economy is one of the largest in the sub-Saharan region, 62.5% of the population lives in rural areas, and 36.8% of the population in 2014 was below

the poverty line. (According to the World Bank, the poverty line is defined as a level of income lower than USD 1.9 per day.) Apart from the demographic situation, in recent years, Kenya's vulnerability has been affected by climate and economic covariate shocks. For example, due to the geographical layout (more than 80% of the available land in Kenya is categorized as arid or semarid land [6]), between 1991 and 2000, Kenya recorded five droughts that affected almost 40 million people due to harsh weather conditions, crop failure, overexploitation of land, and loss of livestock [7]. During one of the most recent droughts (2008/9), it was estimated that 3.5 million people needed immediate food supply, and 6.2 million people were at the starvation level [8]. According to Kenya's Post-Disaster Needs Assessment [9], the economic damage of the 2010/2011 drought was USD 12.1 billion, and the recovery budget should be around USD 1.7 billion. At the time, the Kenyan government requested a transfer of USD 1.2 billion. Hence, droughts are one of the (climate) events that cause agriculture loss, and crop failure imposes a massive economic impact on the Kenyan economy.

In addition to natural disasters, in recent decades, the world economy has been affected by oil price shocks due to political instability [10]. Table 1 shows the events that led to the increase in oil prices, consequently impacting the Kenyan oil sector [11].

**Table 1.** Events in the Kenyan oil sector, 1957–2002 [11].

| Origin of the Shock | Date | Recorded Oil Price Increase |
|---|---|---|
| Suez Crisis | August 1957 | January 1957–February 1957 (9%) |
| OAPEC embargo (Arab–Israel War) | October 1973 | November 1973–February 1974 (51%) |
| Iranian Revolution | January 1980 | May 1979–January 1980 (57%) |
| Iran–Iraq War controls lifted | July 1981 | November 1980–February 1981 (45%) |
| Gulf War I | July 1990 | August 1990–October 1990 (93%) |
| Venezuela unrest, Gulf War II | December 2002 | November 2002–March 2003 (28%) |

According to the World Bank [12], oil prices will rise from USD 56 per barrel in 2021 to USD 70 per barrel in 2035. This evolution will become even more relevant to the Kenyan economy. Indeed, according to the Energy and Petroleum Regulatory Authority [13], the rise of domestic oil consumption (+38% and +164% in terms of oil and liquefied petroleum, respectively) led to a growth in oil import of 53% between 2008 and 2018. Despite the significant improvement in oil production since 2012, Kenya is still a net oil-importing country. For net-importing countries, the increase in oil prices leads to higher production costs and a slowdown in the growth rate, reducing domestic output and increasing inflation [14,15].

Unfortunately, two differentiated shocks may happen in the same period. For instance, oil shocks could occur in the period when the country is suffering from famine because of extreme natural events (e.g., in 2014, Kenya faced an oil shock in conjunction with a drought.).

This study addresses the impact of two contemporaneous covariant shocks (i.e., drought and price oil shock) and possible coping strategies through a static computable general equilibrium (CGE) model applied to Kenyan data. The analysis of these shocks and the coping strategies requires a specific social accounting matrix (SAM) describing the linkages between production, income, consumption, and investment of the agents in the economy. This paper employs a modified version of the SAM for Kenya [16], which is aggregated, focusing on the agricultural sector. Hence, for example, all activities not related to agriculture have been aggregated in several macroactivities (e.g., rest of manufacturing or rest of services), whereas those strictly interconnected with it have been maintained separately (e.g., fertilizers (nitrogen) or water).

The analysis implements a static CGE for two main reasons related to the nature of the shocks. First, it allows for examining both the short-run macroeconomic impact of drought on Kenya's economy and the redistributive consequences of the government's natural disaster mitigation policy without introducing assumptions about the complex climate

dynamics [17,18]. Second, it allows for addressing the short-run effect on a net-importer country of a sudden increase in the oil price, leaving out of the analysis the uncertain evolution of future oil prices.

To the best of the authors' knowledge, this analysis provides an innovative contribution because the effectiveness of a drought mitigation policy is evaluated during a global oil crisis, whereas the CGE literature addresses it separately (see Section 2). Another aspect that makes this study valuable is the attempt to perform scenario analyses as close as possible to the actual situation in Kenya by adopting a static CGE model. In fact, the scenarios are based on Kenyan past events (droughts) or past international oil crises. We therefore adopted a static CGE model because we explore sudden shocks characterized by abrupt but (relatively) short impacts in a developing country with a high level of vulnerability to poverty and with most of the population living in rural areas. Therefore, mitigation policies should be effective in the short term, and a static CGE is an effective tool to evaluate them.

## 2. Literature Review

This study ideally bridges two extensive branches of the CGE modeling literature: (i) the one that addresses the impact of external climate shocks (i.e., natural disasters) and (ii) the one that investigates the effect of oil price shocks, extending the analysis on the possible combined effects on welfare and suggesting some policy responses.

By introducing an agriculture production loss scenario, Ref. [19] investigated the impact of drought and food aid policy in Mozambique. This study shows the positive effects of direct food aid on households during drought. Ref. [20] used a similar method to address the impact of drought on the economy of Botswana and to show the effectiveness of different coping strategies (i.e., external food transfers and food vouchers). In the case of Ethiopia (Awash Basin), Ref. [21] quantified the negative impact of drought on GDP as a reduction of 5% with a more substantial effect on the share of GDP produced by agriculture, which decreased by 10%. Ref. [22] showed the impact of drought on Uganda's economy, highlighting the role of food shortages in economic and welfare losses. In the case of Ethiopia, Ref. [23] showed the negative impact of climate change. Another similar study by [24] on the South African economy showed the unfavorable effects of drought on agriculture. Ref. [8] showed the impacts of extreme weather on Kenya's growth and economy, whereas [25] found comparable results for the economy of Mozambique.

Looking at the CGE literature that addresses the effect of oil shocks, Ref. [26] investigated the impact of oil shocks on the Kenyan economy, highlighting the adverse effects of oil price jumps on domestic institutions and production. Ref. [27] showed the negative impact of soaring oil prices on the welfare of sub-Saharan countries using static and dynamic CGE models. The author used counterfactual simulations for 2002–2008, showing how Kenya lost 2.1% per year in terms of GDP. From studies on oil shocks in other developing countries, Ref. [28] found that oil shocks may devastate welfare but weakly affect poverty in the short run in India. Ref. [29] performed a comparative study on the impact of an oil price shock coupled with implementing a climate policy in Malaysia.

The empirical literature has deeply investigated the impact of oil prices on developing countries. For instance, using an unrestricted vector autoregressive model, Ref. [30] highlighted the adverse effects of oil prices on net-importing developing countries. In contrast, net-exporting countries are more likely to benefit from it. Comparable findings were shown by [31,32]. Focusing on analyses addressing sub-Saharan African countries, Ref. [33] developed an autoregressive distributed lag model highlighting a considerable impact of oil fluctuations on welfare in the case of Ghana. Ref. [11] performed a comparable analysis of the Kenyan economy, discovering that the main transmission channel of the oil shock to the economy in the short run was consumption.

The rest of the paper is structured as follows: Section 3 describes the methodological approach, Section 4 presents the data and the selected exploration scenarios, Section 5 contains the simulation results, and Section 6 suggests concluding remarks.

## 3. Methodology

In line with the relevant literature [34,35], we develop a static computable general equilibrium model to address the impact of two sudden shocks within one period (see the supplementary material in Appendix A for the full description of the model).

Figure 1 illustrates the production process. In the production process, domestic producers maximize their profit subject to their production technology (i.e., a constant elasticity of substitution (CES) function. Production technology includes the factor of production, which will be aggregated with intermediate commodities. Intermediate commodities (raw materials) with a fixed share in the Leontief function contribute to the production of one unit of commodities. The composite commodity basket can provide intermediate commodities. In the composite commodity basket, domestic commodities and imported commodities are aggregated with CES technology. The composite commodities price (market price) includes extra taxes, such as sales tax, tariff tax, and, in some cases, additional market margins.

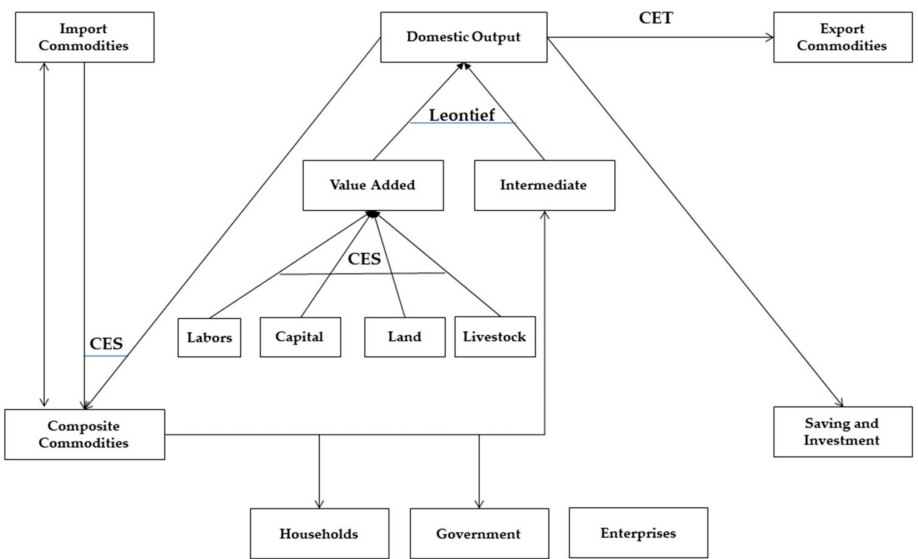

**Figure 1.** The output production process in the CGE model for Kenya.

The number of imports is decided according to the Armington assumption. According to this assumption, the composite commodity production equals domestic demand and is the mixture of domestic and imported goods. Domestic demand is the sum of domestic consumption (household units, enterprises, and government) and the domestic producer's demand (intermediate commodities). Under the baseline assumption, the exchange rate is flexible, and foreign investment (tradeoff between imports and exports in foreign currency) is considered fixed. The savings and investment account is modeled as a fixed share; however, governmental deficit (i.e., savings) is assumed to be flexible to ensure equilibrium in the economy. More details on governmental budget control and market closure can be found in Appendix A.

Households maximize their utility function subject to their budget constraint. Solving the optimization problem, we obtain Equations (1) and (2):

$$PQ_c QH_{c\ h} = PQ_c \gamma^m_{c\ h} + \beta^m_{c\ h}\left(EH_h - \sum_{c'\epsilon C} PQ_{c'}\gamma^m_{c'h} - \sum_{a\epsilon A}\sum_{c'\epsilon C} PXAC_{a\ c'}\gamma^h_{a\ c'h}\right) \quad (1)$$

$$PXAC_{a\ c} QHA_{a\ c\ h} = PXAC_{a\ c}\gamma^h_{a\ c\ h} + \beta^h_{a\ c\ h}\left(EH_h - \sum_{c'\epsilon C} PQ_{c'}\gamma^m_{c'h} - \sum_{a\epsilon A}\sum_{c'\epsilon C} PXAC_{a\ c'}\gamma^h_{a\ c'h}\right) \quad (2)$$

$$EH_h = (1 - \sum_{i\epsilon INSDNG} shii_{i\ h})(1 - MPS_h)(1 - TINS_h)YI_h \quad (3)$$

According to Equation (1), the total value of the household marketed commodity consumption ($PQ_c QH_{c\ h}$) must be equal to the value of the subsistence level of consump-

tion ($PQ_c\gamma^m_{c\ h}$) plus the remaining share of income, called supernumerary income. The subsistence level of consumption is the minimum amount needed for each household, whereas extra consumption depends on the household's income disposal. Gross household income is from factor endowment and transfers from other institutions. Due to the lack of information, we assumed a constant endowment factor. Rental price and wages are endogenous in the baseline version of the model. Equation (3) shows that household net income ($EH_h$) is equal to gross income ($YI_h$) minus the direct tax rate paid to the government ($TINS_h$) and the marginal propensity of the saving rate ($MPS_h$) paid to the savings and investment account. The household's gross income is also reduced by transfers to other domestic nongovernmental institutions with the share of $shii_{i\ h}$. Equation (2) represents the case where households have self-consumption (i.e., households themselves produce goods that they consume). This concept is common to the African economy. The price for these commodities has no tax and margin, which is the difference between the values of consumption in Equation (2) ($PXAC_{a\ c} \times QHA_{a\ c\ h}$) and Equation (1).

Since we do not have a credit market, the income of each institution should be equal to its expenditure. The subsistence level of consumption should be estimated based on the Fischer parameter [36] for each household type.

$$w_h = -\frac{EH_h}{EH_h - PQ_c\gamma^m_{c\ h}} \tag{4}$$

The Fischer parameter ($w_h$) is the income elasticity of the marginal utility of income, which can be seen in Equation (4). This parameter is needed to estimate the unknown subsistence level of consumption. To assess the Fischer parameter, we can refer to Equation (5) according to the method developed by [37]:

$$-w \approx 36X^{-3.6} \tag{5}$$

where $w$ is the Fischer parameter and $X$ is the GNP per capita in US dollar exchanged to Kenyan shilling (KES).

Estimating this parameter is essential for our analysis to get a more accurate picture of the different consumption patterns of varying household quintiles (from rich to poor). As Equation (5) shows, the richer the household, the smaller the Fischer parameter number in absolute value.

From the demand side, the factor market includes domestic and international demanders, and from the supply side, factor owners, such as households and enterprises. Wages can fluctuate according to demand when supply is constant. As with any other market in the CGE model, factor supply must equal factor demand when in equilibrium. Elasticities in this study were derived from estimates made by [38] based on an econometric approach.

## 4. Data and Simulations

To identify the linkages between drought, agricultural production, food price changes, and welfare losses, we developed a modified version of the 2014 Kenyan social accounting matrix (SAM). The details and specifications of the 195 accounts of the original SAM can be found in [16]. A social account matrix is a comprehensive, economy-wide database reporting the value of all transactions (in this case, expressed in Kenyan shilling) among agents over a period of time, usually 1 year. The new SAM includes 22 activities and 20 marketed goods. The modified SAM takes into account the dual role of households as producers and consumers. This requires separating the production inputs of these households according to their destination: self-consumption and market [39].

Rural household duality and drought are crucial for the agricultural sector. For this reason, the SAM has been aggregated from an agrocentric perspective. SAM classifies agricultural activities into six accounts: coffee, which is the cash crop in Kenya; aggregate food crops; dairy (meat, dairy, and livestock); the rest of agriculture, and two representing households as food- and cash-crop-producing activities.

Households are aggregated into 12 accounts: 5 accounts for each of Kenya's two major cities (Nairobi and Mombasa), dividing households according to their income in the quintile distribution (from Q1 to Q5), and 2 for the rest of Kenya (rural and urban—without Nairobi and Mombasa). This classification allows us to perform a redistributive analysis of covariate shocks and mitigation policies.

The factors of production are land (irrigated and nonirrigated), capital (agricultural and nonagricultural), livestock, and labor. The labor factor is disaggregated into high-skilled, semiskilled, and unskilled labor, regionalized into four locations (Nairobi, Mombasa, nonirrigated rain-fed areas, and the rest of Kenya).

As a result, the new Kenyan SAM structure has 88 accounts. Table 2 shows the revised SAM with details and the number of accounts for each group (see Appendix A for further description of activities and accounts).

**Table 2.** Modified Kenya SAM 2014.

| Accounts | Details | Numbers |
| --- | --- | --- |
| Activities | Activities | 20 |
| | Household as an activity | 2 |
| Commodities | Marketed commodities | 20 |
| Factors of production | Labors | 12 |
| | Capital | 2 |
| | Land | 2 |
| | Livestock | 1 |
| Domestic nongovernmental institutions | Households | 12 |
| | Enterprises | 1 |
| Government | Government | 1 |
| Taxes | Direct tax | 1 |
| | Indirect tax | 1 |
| | Factor tax | 1 |
| | Sales tax | 1 |
| | Tariff on rest of the world | 1 |
| Rest of the world | Rest of the world | 1 |
| Saving and investment | Investment | 1 |

To capture the economic impact of oil and drought shocks, three main scenarios were designed: scenario A (drought without policy response), scenario B (drought with policy response), and scenario C (drought with policy response and oil shock).

Scenario A is a Hicks-neutral technological shock [22,24]. A shock is defined as Hicks-neutral if it does not change the balance of factors of production in the production function [40]. The Hicks-neutral technological shock will affect the efficiency parameter in the production function, considering a fixed amount of factor supply. We assume that the climate change shock (i.e., drought) reduces the efficiency parameter $\alpha_a^{va}$ of Equation (6):

$$QVA_a = \alpha_a^{va} \left( \sum_{f \epsilon F} \delta_{f\,a}^{va} QF_{f\,a}^{-\rho_a^{va}} \right)^{\frac{1}{\rho_a^{va}}} \qquad (6)$$

where $QVA_a$ is the quantity of value added for activity $a$, $QF_{f\,a}$ is the quantity of factor $f$ being demanded in activity $a$, $\alpha_a^{va}$ is the efficiency parameter for activity $a$, $\delta_{f\,a}^{va}$ is the share of factor $f$ used in activity $a$, and $\rho_a^{va}$ is a value-added function exponent. In line with the reference literature [21,41–43], the amount of average reduction across all agriculture commodities for this scenario is 10%. (See in Table A6 in Appendix B a comparison with [19] on the Hicks-neutral technological shock.)

In the scenario with drought and coping strategy (scenario B), we reproduce the strategy applied by the Kenyan government in response to previous droughts [44,45] (i.e., transfers to households). During past drought crises, the government applied programs such as the Hunger Safety Net Program [46] or received funds in the form of drought contingency programs [47] from external donors and funders, such as the United States Agency for International Development (USAID), the International Cooperation and Development at the European Commission (DEVCO), and the European Civil Protection and Humanitarian Operations (ECHO).

The amount of funds and transfers from/to the Kenyan government could be derived from evidence in the past. During the 2011 drought, the Kenyan government requested a USD 1.2 billion fund, from which USD 969 million was received by the government of Kenya from international organizations and donors. For the sake of simplicity, we assume that the requested funds are received over 1 year and that the government reallocates the requested transfers to households across the country based on the amount of damage suffered. Hence, the amounts of transfers are proportional to the welfare loss of each household category. The last scenario (scenario C) combines scenario B with an increase in the world price of oil. To evaluate the impact of the oil price, we perform some sensitivity analysis on oil price shock, exploring the effects of 5%, 10%, and 20% increases in oil prices.

Following the [12], the oil shocks are modeled as a percentage increase in world oil price as follows:

$$PM_{Oil} = Pwm_{Oil}(1 + tm_{Oil})EXR + \sum_{c' \epsilon CT} PQ_{c'}icm_{c'\ Oil} \tag{7}$$

In this equation, the import price ($PM_{Oil}$) is equal to the world oil price ($Pwm_{Oil}$) in foreign currency (US dollar) multiplied by the tariff tax rate parameter ($tm_{Oil}$) and exchange rate (EXR) plus the value of import margin ($PQ_{c'}icm_{c'\ Oil}$). World oil price in Equation (7) is subjected to a 5%–10%–20% change from the base value.

## 5. Simulation Results

In this section, by examining the results of drought without policy response (scenario A) and drought with policy response (scenario B), we discuss the effectiveness of the policy implemented during the drought. Then, we analyze the results of the scenario with drought, policy response, and oil shock (scenario C), comparing them with scenario B. By performing this comparative study, we can evaluate the policy response to a widespread drought with or without an oil price shock (scenario A (10% reduction in agriculture output) in detail could be seen in Appendix B).

### 5.1. Drought Mitigation Policy Effectiveness

Based on the mitigation strategies developed in the past by the Kenyan government to handle drought consequences [44,45], this section explores the impact of extending fund transfers to households. Scenario B assumes that the Kenyan government receives financial aid of KES 130 billion (USD 1.16 billion) and transfers to households based on their expenditure losses. This assumption is based on the previous drought mitigation policy implemented by the Kenyan government under drought contingency and the Hunger Safety Net Program for the drought in 2011. Under these programs, the government of Kenya requested USD 1.2 billion from international donors to deal with the negative impact of drought between 2009 and 2011 [46,47]. In the past, the government requested and received the funds based on the estimated damages. To be consistent with this procedure, we assume that the government requests funds based on the economic losses (i.e., damages) due to the reduction in agriculture productivity. Therefore, it is possible to read Table 3 from two perspectives: (i) it shows the losses in monetary terms from the households' point of view, and (ii) it exhibits the number of transfers from the government to the household units. Given the demographic distribution of the Kenyan population, the amount of transfers (i.e., damage suffered) to rural areas is about 58% of the whole available funds of the Kenyan government.

**Table 3.** The number of transfers equals the expenditure loss from the government to the households (scenario A).

| Household Units | Transfers from the Government (Billion KES) |
|---|---|
| Nairobi Quantile 1 (Richest) | 13.821 |
| Nairobi Quantile 2 | 4.93 |
| Nairobi Quantile 3 | 3.055 |
| Nairobi Quantile 4 | 2.231 |
| Nairobi Quantile 5 (Poorest) | 0.648 |
| Mombasa Quantile 1 (Richest) | 2.094 |
| Mombasa Quantile 2 | 1.363 |
| Mombasa Quantile 3 | 0.82 |
| Mombasa Quantile 4 | 0.48 |
| Mombasa Quantile 5 (Poorest) | 0.217 |
| Rest of Kenya Urban | 24.218 |
| Rest of Kenya Rural | 75.258 |
| Total | 129.135 |

Figures 2 and 3 show the impact of the mitigation policy on households' income and equivalent variation (EV). Equivalent variation is an index for welfare change due to the consumption price changes, hence, to utility changes. Public transfers increase the total welfare of the households in scenario B (+3.2% than the scenario without policy response). The mitigation policy has redistributive effects on both income and welfare. Indeed, excluding the two major cities, income and welfare in urban areas increased by 4.72% and 3.5%, respectively, whereas in rural communities, income and welfare rose by 3.95% and 3.5%, respectively. Both Figures show improvements in household income and welfare. However, Figure 3 shows that the rural area registers falls in terms of welfare, despite the highest share of transfers from the government to this area (see Table 3). This is mainly because the consumption basket in rural areas highly relies on agrifood commodities, and these areas are most populated compared with urban areas.

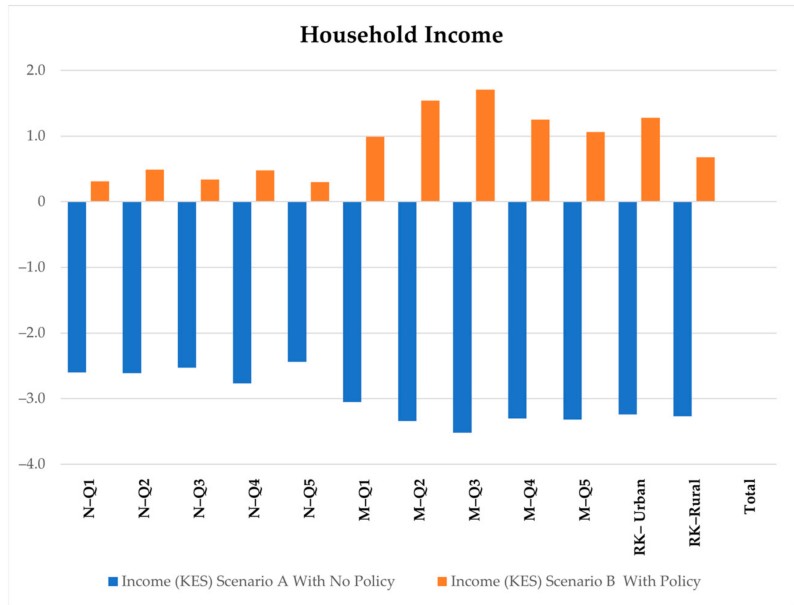

**Figure 2.** Percentage change in household's income. Scenario A (drought) in blue and scenario B (drought with policy response) in orange. N means Nairobi, M refers to Mombasa, whereas RK describes the rest of Kenya. The households in Nairobi and Mombasa are divided in quantile, from the richest (Q1) to the poorest (Q5). (For interpretation of the references to color in the figure legend, the reader is referred to the web version of this article.).

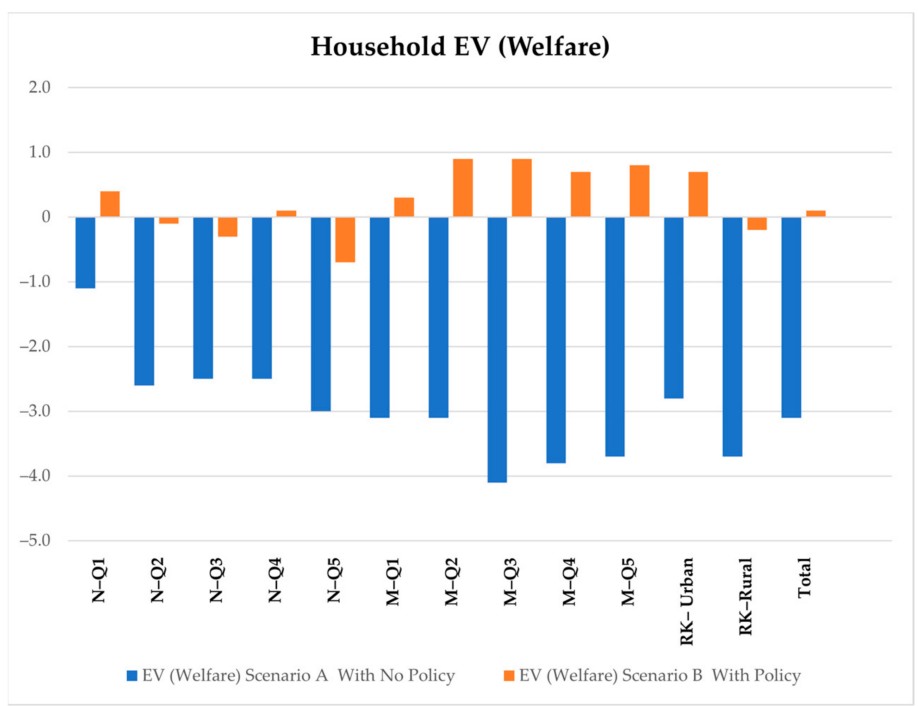

**Figure 3.** Percentage change in household's EV (welfare). Scenario A (drought) in blue and scenario B (drought with policy response) in orange. N means Nairobi, M refers to Mombasa, whereas RK describes the rest of Kenya. The households in Nairobi and Mombasa are divided in quantile, from the richest (Q1) to the poorest (Q5).

Table 4 shows the national accounts and impact of drought with or without mitigation policy on indicators, such as total absorption, private consumption, export and import, exchange rate, and GDP. Looking at the aggregate impact of the mitigation policy, Table 4 shows that the total absorption in scenario A shrinks by −2.12%, whereas the policy can fully counterbalance the drought effect (+0.06%). Public transfers positively affect private consumption, which in scenario B increases by +0.09%, whereas in the no-policy scenario, it decreases by −3.05%. The absence/presence of the policy response has substantial effects on the trade balance. In scenario A, the increase in the exchange rate (0.93%) combined with the reduction in household income contracts for both exports (−2.59%) and imports (−1.36%). On the contrary, in scenario B, the Kenyan shilling appreciates by +5.24%, boosting the import (+2.34%) and shrinking the export by −9.20%, worsening the trade balance. The mitigation can only partially offset the negative impact of drought, but it allows for maintaining a higher GDP level than the no-policy scenario (+0.12%).

**Table 4.** Percentage change in real GDP and national accounts, scenario A and B.

| Real GDP and National Accounts | Drought Shock | |
|---|---|---|
| | Scenario A, without Policy | Scenario B, with Policy |
| Absorption | −2.12 | 0.06 |
| Private consumption | −3.05 | 0.09 |
| Exports | −2.59 | −9.20 |
| Imports | −1.36 | 2.34 |
| GDP (at market prices) | −2.47 | −2.35 |
| Exchange rate | 0.93 | −5.24 |

### 5.2. Oil Shock in the Middle of a Climate Crisis

We conclude our analyses by investigating some scenarios in which, while the government is managing a climate crisis, the economy is hit by an exogenous oil shock, leading to an increase in the oil price. These scenarios are consistent with the double shocks that the Kenyan economy faced in 2014 when the oil price dropped sharply in the second part of the year during a severe drought.

Figures 4 and 5 show the impact of the oil shock in terms of market prices and domestic sales in a scenario where the government is facing a drought. Compared with scenario B (i.e., drought mitigation policy without oil shock), the shocks strongly affect the petroleum price, which increases by 4.45%, 8.88%, and 17.86%, respectively, in the presence of 5%, 10%, and 20% oil shocks. The agricultural industries, particularly those most affected by the drought, register an increase in market prices and a decrease in demand due to production deficiencies. However, by increasing the oil shock (from 0% to 20%), we can see a decreasing trend in agricultural market prices (Figure 4) and an increase in domestic demand (Figure 5). For instance, the growth in the food crops price reduces from +1.85% (in scenario B, i.e., no oil shock) to +1.20% in scenario C with a 20% oil shock. For most of the manufacturing industries, except for "textile and clothing" and "leather and footwear", the oil shocks increase domestic demands compared with scenario B. The most robust oil price shock produces more pronounced effects in the services industry than in agriculture or manufacturing. Indeed, market prices and domestic sales in services show a downward trend with the increase in oil prices. By increasing the oil prices (i.e., oil shock), the agriculture market prices (such as "food crops" and "meat–dairy–livestock") show a downward trend, whereas the manufacturing market prices and domestic sales exhibit an increasing trend, as a result of changes in export and import demand. Since Kenya is a net exporter in agrifood industries, a reduction in domestic demand coupled with decreasing market prices during an oil shock can make the export more profitable. To verify this result, we can look at the results of the simulations regarding the trade balance.

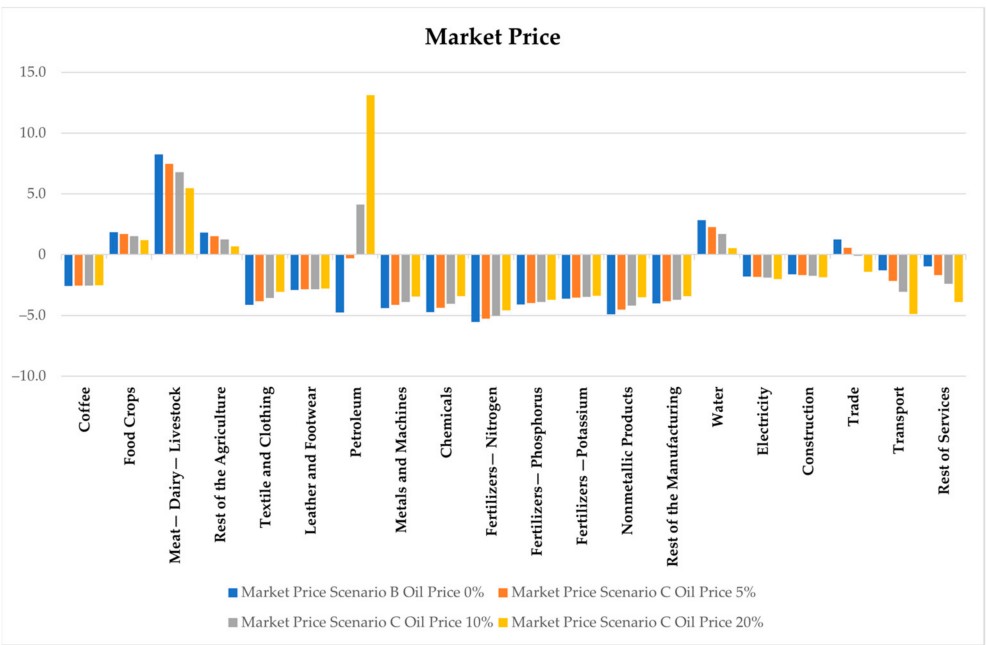

**Figure 4.** Percentage change in market prices, scenario B (drought with policy response) and scenario C (drought with policy and oil shock).

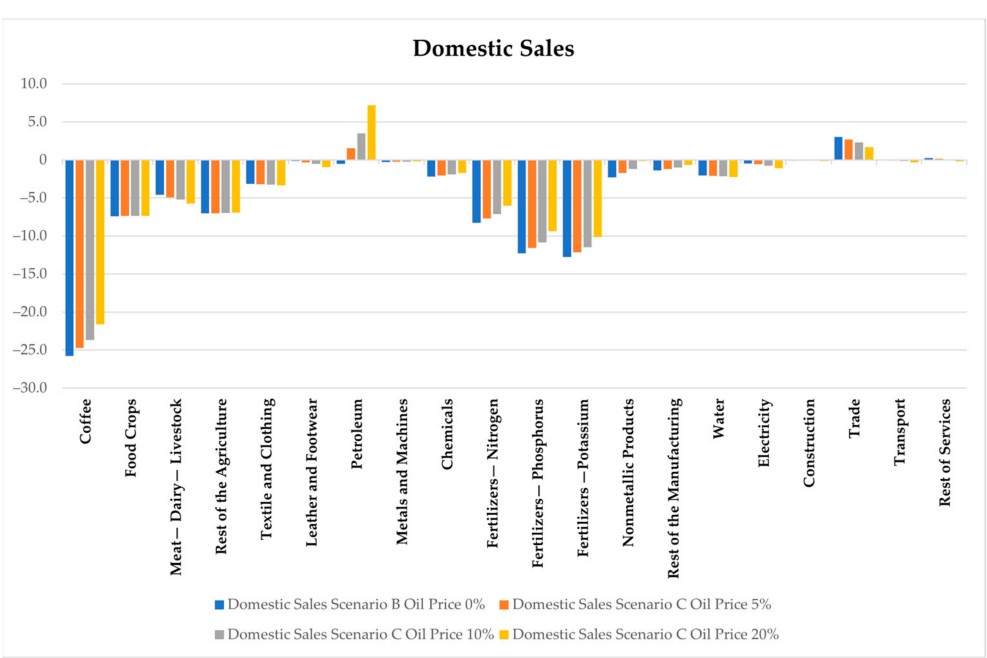

**Figure 5.** Percentage change in domestic sales, scenario B (drought with policy response) and scenario C (drought with policy and oil shock).

Table 5 shows the drought policy's impact on trade with and without oil price shocks. In terms of imports of agricultural products, the scenario without oil shock (scenario B) shows a more significant increase for most productions than the scenario with a double shock. As the oil price increases, agricultural commodities show a decreasing trend in imports and an increasing trend in exports. On the one hand, these dynamics are in line with the reduction of the agricultural, domestic sales for agricultural commodities (see Figure 5). The increase in the price of oil and raw materials used in the production process negatively affects the purchasing power of the households (particularly, fertilizers for agriculture production show the increasing prices with oil shocks in Table 5). However, the growth in the price of these intermediate goods leads to an increase in the price of agricultural commodities, positively affecting the profitability of agriculture commodity export during the oil shock. Moreover, the increase in the world oil price reduces petroleum imports and most manufacturing commodities. Finally, fertilizers registered the highest reduction in either import or export quantity regardless of oil prices. This may be due to the agricultural industry's significant decrease in production during the drought shock. The trend of increasing exports of agricultural commodities can be explained by the development of domestic sales and market prices (see Figures 4 and 5). While domestic demand for agrifood products decreases during the oil shock due to the reduction in the purchasing power of domestic consumers, the reduction in domestic product prices may lead to an increase in export products. Therefore, domestic suppliers are more likely to export their products in order to sell them to the external market at relatively higher prices than in the domestic market.

**Table 5.** Percentage change in import and export prices.

| Marketed Commodities | Drought Shock Policy and Oil Shock | | | | | | | |
|---|---|---|---|---|---|---|---|---|
| | Import Quantity | | | | Export Quantity | | | |
| | Scenario B | | Scenario C | | Scenario B | | Scenario C | |
| | Oil Price +0% | Oil Price +5% | Oil Price +10% | Oil Price +20% | Oil Price 0% | Oil Price +5% | Oil Price +10% | Oil Price +20% |
| Coffee | −22.72 | −21.71 | −20.73 | −18.73 | −29.27 | −27.83 | −26.42 | −23.53 |
| Food crops | 15.96 | 14.33 | 12.77 | 9.74 | −17.30 | −16.50 | −15.73 | −14.18 |
| Meat–dairy–livestock | 19.84 | 19.02 | 18.46 | 17.53 | −19.69 | −18.75 | −17.88 | −16.14 |
| Rest of the agriculture | 9.90 | 9.13 | 8.39 | 6.91 | −16.84 | −15.94 | −15.07 | −13.31 |
| Textile and clothing | 1.84 | 1.22 | 0.60 | −0.65 | −10.87 | −9.94 | −9.07 | −7.29 |
| Leather and footwear | −0.18 | −0.37 | −0.60 | −1.04 | −4.88 | −4.21 | −3.58 | −2.38 |
| Petroleum | 1.05 | −0.83 | −2.59 | −5.61 | −3.20 | −4.98 | −6.82 | −10.62 |
| Metals and machines | −2.69 | −2.32 | −1.99 | −1.36 | −1.46 | −1.10 | −0.75 | −0.02 |
| Chemicals | 0.48 | 0.04 | −0.41 | −1.35 | −5.98 | −5.03 | −4.13 | −2.27 |
| Fertilizers—nitrogen | −11.18 | −11.17 | −11.16 | −11.16 | −5.01 | −3.51 | −2.03 | 1.05 |
| Fertilizers—phosphorus | −10.65 | −10.69 | −10.73 | −10.82 | −15.59 | −13.68 | −11.78 | −7.80 |
| Fertilizers—potassium | −10.40 | −10.43 | −10.47 | −10.55 | −17.34 | −15.55 | −13.79 | −10.09 |
| Nonmetallic products | −1.01 | −0.89 | −0.78 | −0.58 | −4.27 | −3.11 | −1.97 | 0.40 |
| Rest of manufacturing | 1.03 | 0.57 | 0.11 | −0.82 | −5.40 | −4.22 | −3.07 | −0.70 |
| Water | 0.00 | 0.00 | 0.00 | 0.00 | 0.00 | 0.00 | 0.00 | 0.00 |
| Electricity | 0.00 | 0.00 | 0.00 | 0.00 | 0.00 | 0.00 | 0.00 | 0.00 |
| Construction | 0.00 | 0.00 | 0.00 | 0.00 | 0.00 | 0.00 | 0.00 | 0.00 |
| Trade | 6.53 | 5.53 | 4.60 | 2.79 | −0.35 | −0.11 | 0.11 | 0.60 |
| Transport | 2.13 | 1.34 | 0.56 | −1.05 | −2.13 | −1.49 | −0.86 | 0.43 |
| Rest of services | 2.53 | 1.80 | 1.09 | −0.37 | −1.97 | −1.46 | −0.97 | 0.02 |

Figures 6 and 7 show the impact of oil shocks on the effectiveness of the drought mitigation policy in terms of household's income and EV (welfare). In terms of household's income (Figure 6), the oil shock erodes the positive effects of the drought mitigation policy (scenario B). Nevertheless, with a +5% oil price shock, other parts of the country still can benefit from the drought mitigation policy except for Nairobi household units. The reduction in household income in Nairobi is mainly because, according to the data, workers from Nairobi are employed more in the oil-related industries, such as those that deal with petroleum production, chemicals, and fertilizers. Looking at the income changes in the two main cities, Figure 6 shows that the average reduction in the first two richest quintiles (Q1 and Q2) is lower than in the two lowest income quantiles (Q4 and Q5). These findings confirm the results of scenario B, according to which richer quantiles experience more remarkable improvement due to government transfers. These results can be explained by the mitigation policy design, which allocates many transfers to the first two quantiles in both cities. Having a higher income/wealth level, richer quantiles register more significant losses in absolute terms due to drought. Then, according to the mitigation policy, they will receive more transfers. This will result in lower losses for these quantiles than for poorer ones, with regressive consequences on income/wealth distribution. Comparing the results between urban and rural areas in both scenarios, even if rural communities receive a higher number of transfers, they will register lower net benefits than urban communities. All these results are confirmed in sign and even more so in magnitude regarding welfare (Figure 7).

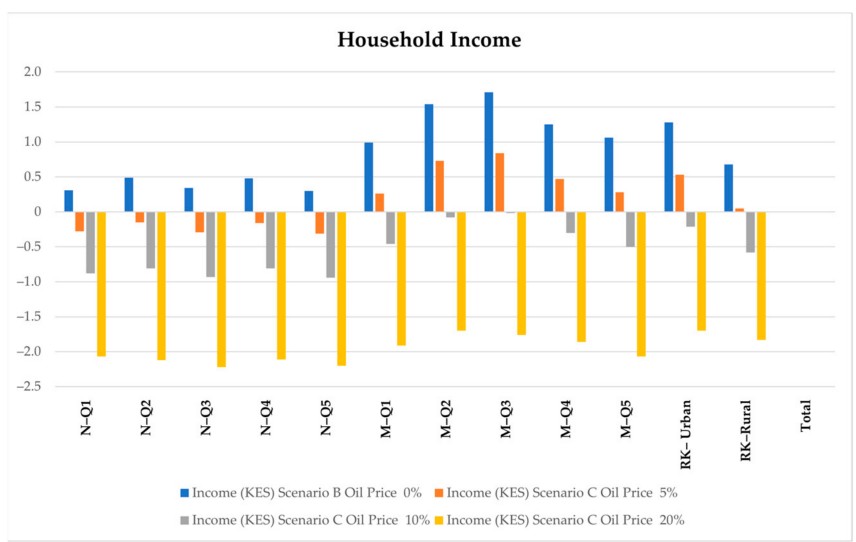

**Figure 6.** Percentage change in household income, scenario B (drought with policy response) and scenario C (drought with policy and oil shock). N means Nairobi, M refers to Mombasa, whereas RK describes the rest of Kenya. The households in Nairobi and Mombasa are divided in quantile, from the richest (Q1) to the poorest (Q5).

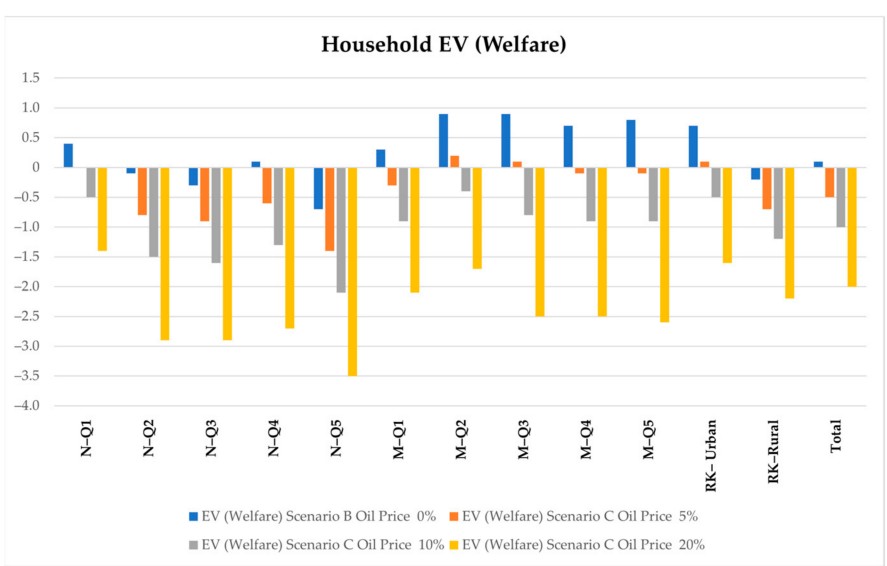

**Figure 7.** Percentage change in household's EV (welfare), scenario B (drought with policy response) and scenario C (drought with policy and oil shock). N means Nairobi, M refers to Mombasa, whereas RK describes the rest of Kenya. The households in Nairobi and Mombasa are divided in quantile, from the richest (Q1) to the poorest (Q5).

Table 6 contains total absorption, private consumption, overall export and import, exchange rate, and GDP. This table shows the negative impact of the oil shock on the main aggregate variables. For example, the most potent oil shock reduces the total absorption and private consumption by 1.46% and 2.09%, compared with the scenario with no oil price increase. In terms of trade, the imports show a reduction in the oil price shock, from a growth of 2.34% in the scenario without shock to a decrease of −0.77% in the worst scenario C (+20% oil price). On the other hand, exports increased by 2.77% in the +20% oil price shock scenario compared to scenario B.

**Table 6.** Percentage change in real GDP and national accounts, scenarios B and C.

| Real GDP and National Accounts | Drought Shock Policy and Oil Shock | | | |
| --- | --- | --- | --- | --- |
| | Scenario B | Scenario C | | |
| | Oil Price +0% | Oil Price +5% | Oil Price +10% | Oil Price +20% |
| Absorption | 0.06 | −0.31 | −0.68 | −1.40 |
| Private consumption | 0.09 | −0.45 | −0.98 | −2.00 |
| Exports | −9.20 | −8.49 | −7.81 | −6.43 |
| Imports | 2.34 | 1.50 | 0.70 | −0.77 |
| GDP (at market prices) | −2.35 | −2.38 | −2.42 | −2.50 |
| Exchange rate | −5.24 | −4.78 | −4.36 | −3.5 |

Finally, the oil price shock negatively affects the GDP, which in the worst scenario registers a further decline by 7.959 billion KES (−0.15%) compared to scenario B.

## 6. Conclusions and Remarks

Global climate change and the continuous depletion of nonrenewable natural resources may lead to a future characterized by a higher frequency of extreme climate events (i.e., floods, hurricanes, or droughts) and resource supply shocks (i.e., oil price shocks). Sub-Saharan African countries are already particularly exposed to these types of shocks because of both their socioeconomic conditions and geographical conformation.

This article presents a static CGE model applied to Kenya to investigate the effectiveness of a mitigation policy (transfers to the population) in managing a climate crisis (i.e., a drought) whose effect may be exacerbated by a contemporaneous exogenous oil price shock. This scenario is coherent with what happened in 2014 when the price of oil dropped sharply by 30% during a severe drought.

To address the redistributive implications of the shocks and the mitigation policy, we estimate the Fisher parameter [36] for each household quantile using the method introduced by [37]. We assume a range of oil shocks that lead to a jump in world oil prices by different magnitudes (+5%, +10%, and +20%). In contrast, the drought strongly affects agricultural production, shrinking the average output by 10%. Based on the past postdrought strategies of the Kenyan government, the climate crisis mitigation policy implemented by the government consists of externally financed transfers to households.

The analysis shows that public transfers are an effective tool for mitigating drought effects, improving welfare and GDP in the short run (+3.2% and +0.12%, respectively) compared with the scenario without postdrought policy. Looking at the effectiveness of the mitigation policy in terms of income variation at the household level, the model suggests that transfers positively affect both the two richer quantiles and the middle-lower quantiles with comparable magnitude in the two main cities. In contrast, they have a more substantial impact on urban than rural communities.

However, implementing this mitigation policy to manage drought during an oil crisis may have regressive effects on the population and negatively impact the overall economy. Results of the analysis suggest that, in the domestic market, an increasing trend in oil prices with existing drought mitigation policy leads to a decreasing trend in market prices and a rising trend in domestic sales for agriculture and manufacturing industries. In terms of trade, imports show a downward trend in agriculture compared with an upward trend in exports. Considering that Kenya is a net exporter of agricultural commodities, this means that despite more expensive oil prices, exports can become more profitable for agriculture industries.

Regarding household income and welfare, our findings suggest a regressive impact of oil shocks on drought mitigation policy. Indeed, comparing the two extreme quantiles (richest and poorest), the wealthiest quantile always registers a lower reduction in income and welfare than the poorest quantile when a climate crisis is coupled with an oil shock.

The explanation of this result is the design of the mitigation policy, which allocates a higher number of transfers from the government to the richer quantiles since they register a more substantial reduction in expenditure (in absolute terms).

In conclusion, mitigation policy effectiveness can be strongly affected by the complex interaction between drought and oil shocks, and public authorities should pay attention to this in policy implementation. Our study shows that the complexity of policy implementation is highly related to the levels of contemporaneous shocks. On the other hand, these findings call for a new scheme of transfer allocation, in which rural and low-income household quintiles should receive more attention from mitigation policy in the postdrought period, as they are the most vulnerable to external shocks.

**Author Contributions:** Conceptualization, D.B., A.M. and S.V.; Data curation, A.M.; Formal analysis, A.M.; Investigation, A.M.; Methodology, A.M.; Software, A.M.; Supervision, D.B. and S.V.; Visualization, D.B., A.M. and S.V.; Writing—original draft, D.B., A.M. and S.V.; Writing—review & editing, D.B., A.M. and S.V. All authors have read and agreed to the published version of the manuscript.

**Funding:** This research received no external funding.

**Conflicts of Interest:** The authors declare no conflict of interest.

## Appendix A. Model Specification

**Table A1.** Sets.

| Sets | Description | Sets | Description |
|------|-------------|------|-------------|
| A | Activities | CM | Commodities with import |
| C | Commodities | CMN | Commodities without import |
| CD | Commodities with domestic sales | CT | Transaction service commodities |
| CDN | Commodities without domestic sales | CX | Commodities with domestic product |
| CE | Commodities with export | F | Factors |
| CEN | Commodities without export | INS | Institutions |
| INSD | Domestic institutions | INSDNG | Domestic nongovernmental institutions |
| H | Households | ALEO | Activities with Leontief technology |

**Table A2.** Parameters.

| Sets | Description | Sets | Description |
|------|-------------|------|-------------|
| $cwts_c$ | Weight of commodity c in CPI | $\overline{qg_c}$ | Base-year quantity of government demand |
| $ica_{c\,a}$ | Input–output coefficient | $\overline{qinv_c}$ | Base-year quantity of private investment demand |
| $ice_{c'\,c}$ | Share of trade cost in export | $shift_{i\,f}$ | Share for domestic institution i in the income of factor f |
| $icm_{c'\,c}$ | Share of trade cost in import | $shii_{i\,i'}$ | Share of net income of i to i |
| $\overline{mps_i}$ | Base saving rate of domestic institutions i | $ta_a$ | Tax rate for activity a |
| $mps01_i$ | 0–1 parameter with 1 for institutions with potentially flexed direct tax rates | $te_c$ | Export tax rate |
| $pwe_c$ | Export price (foreign currency) | $tf_f$ | Direct tax rate |
| $pwm_c$ | Import price (foreign currency) | $\overline{\overline{tins_i}}$ | Exogenous direct tax rate for domestic institution i |
| $tins01_i$ | 0–1 parameter with 1 for institutions with potentially flexed direct tax rates | $tm_c$ | Import tariff rate |
| $tq_c$ | Rate of sales tax | $trnsfr_{i\,f}$ | Transfer from factor f to institution i |
| $tva_a$ | Rate of value-added tax for activity a | | |

**Table A3.** Exogenous Variables.

| Variable | Description | Variable | Description |
|---|---|---|---|
| $\overline{CPI}$ | Consumer price index | $\overline{MPSDJ}$ | Savings rate scaling factor (=0 for base) |
| $\overline{DTINS}$ | Change in domestic institution tax share (=0 for base; exogenous variable) | $\overline{QFS_f}$ | Quantity supplied of factor |
| $\overline{FSAV}$ | Foreign savings (FCU) | $\overline{TIASNDJ}$ | Direct tax scaling factor (=0 for base; exogenous variable) |
| $\overline{GADJ}$ | Government consumption adjustment factor | $\overline{FEND_{f\,a}}$ | Wage distortion factor for factor f in activity a |
| $\overline{IADJ}$ | Investment adjustment factor | | |

**Table A4.** Endogenous Variables.

| Variable | Description | Variable | Description |
|---|---|---|---|
| DMPS | Change in domestic institution savings rates (=0 for base; exogenous variable) | $QH_{c\,h}$ | Quantity consumed of commodity c by household h |
| EG | Government expenditures | $QHA_{a\,c\,h}$ | Quantity of household home consumption of commodity c from activity a for household h |
| $EH_h$ | Consumption spending for household | $QINTA_a$ | Quantity of aggregate intermediate input |
| EXR | Exchange rate (LCU per unit of FCU) | $QINT_{c\,a}$ | Quantity of commodity c as intermediate input to activity a |
| GOVSHR | government consumption shares in nominal absorption | $QINV_c$ | Quantity of investment demand for the commodity |
| GSAV | Government savings | $QM_c$ | Quantity of imports of the commodity |
| INVSHR | Investment share in nominal absorption | $QQ_c$ | Quantity of goods supplied to the domestic market (composite supply) |
| $MPS_i$ | Marginal propensity to save for domestic nongovernmental institutions (exogenous variable) | $QT_c$ | Quantity of a commodity demanded as a trade input |
| $PA_a$ | Activity price (unit gross revenue) | $QVA_a$ | Quantity of (aggregate) value-added |
| $PDD_c$ | Demand price for commodities produced and sold domestically | $QX_c$ | Aggregated marketed quantity of domestic output of commodity |
| $PDS_c$ | Supply price for commodities produced and sold domestically | $QXAC_{a\,c}$ | Quantity of marketed output of commodity c from activity a |
| $PE_c$ | Export price (domestic currency) | TABS | Total nominal absorption |
| $PINTA_a$ | Aggregate intermediate input price for activity a | $TINS_i$ | Direct tax rate for institution i |
| $PM_c$ | Import price (domestic currency) | $TRII_{i\,i'}$ | Transfers from institution i. to i (both in the set INSDNG) |
| $PQ_c$ | Composite commodity price | $WF_f$ | Average price of factor f |
| $PVA_a$ | Value-added price (factor income per unit of activity) | $YF_f$ | Income of factor f |
| $PX_c$ | Aggregate producer price for the commodity | YG | Government revenue |
| $PXAC_{a\,c}$ | Producer price of commodity c for activity a | $YI_i$ | Income of domestic nongovernmental institution |
| $QA_a$ | Quantity (level) of activity | $YIF_{i\,f}$ | Income to domestic institution i from factor f |
| $QD_c$ | Quantity sold domestically of domestic output | | |
| $QE_c$ | Quantity of exports | | |

**Table A4.** *Cont.*

| Variable | Description | Variable | Description |
|----------|-------------|----------|-------------|
| $QF_{f\,a}$ | Quantity demanded of factor f from activity a | | |
| $QG_c$ | Government consumption demand for the commodity | | |

**Table A5.** List of Aggregated SAM Accounts.

| Activities | Commodities |
|------------|-------------|
| Coffee | |
| Food crops | Coffee |
| Dairy (meat, dairy, livestock) | Food crops |
| Rest of agriculture | Dairy (meat, dairy, livestock) |
| Household as an activity food | Rest of agriculture |
| Household as an activity cash crop | |
| Chemicals | Chemicals |
| Textile and clothing | Textile and clothing |
| Leather and footwear | Leather and footwear |
| Petroleum | Petroleum |
| Metals and machines | Metals and machines |
| Fertilizers—nitrogen | Fertilizers—nitrogen |
| Fertilizers—phosphorous | Fertilizers—phosphorous |
| Fertilizers—potassium | Fertilizers—potassium |
| Nonmetallic product | Nonmetallic product |
| Rest of the manufacturing | Rest of the manufacturing |
| Water | Water |
| Electricity | Electricity |
| Construction | Construction |
| Trade | Trade |
| Transport | Transport |
| Rest of services | Rest of services |

| Factors of Production | Institution Account |
|-----------------------|---------------------|
| Labor—skilled Nairobi | |
| Labor—semiskilled Nairobi | Nairobi—quintile 1 (richest) |
| Labor—unskilled Nairobi | Nairobi—quintile 2 |
| Labor—skilled Mombasa | Nairobi—quintile 3 |
| Labor—semiskilled Mombasa | Nairobi—quintile 4 |
| Labor—unskilled Mombasa | Nairobi—quintile 5 (poorest) |
| Labor—skilled rainfall | Mombasa—quintile 1 (richest) |
| Labor—semiskilled rainfall | Mombasa—quintile 2 |
| Labor—unskilled rainfall | Mombasa—quintile 3 |
| Labor—skilled rest of Kenya | Mombasa—quintile 4 |
| Labor—semiskilled rest of Kenya | Mombasa—quintile 5 (poorest) |
| Labor—unskilled rest of Kenya | Rest of Kenya—rural |
| Land irrigated | Rest of Kenya—urban |
| Land nonirrigated | Government |
| Livestock | Enterprises |
| Capital agricultural | Rest of the world |
| Capital nonagricultural | |

| Tax Accounts | Other Accounts |
|--------------|----------------|
| Factor tax | |
| Direct tax | |
| Indirect tax | Investment |
| Sales tax | |
| Tariffs on the rest of the world | |

*Appendix A.1. Activities*

Activities across the economy maximize their profit subject to their budget constraint, that is, the cost of using varied production factors with the CES function technology (constant elasticity of substitution). Producers use composite commodities (including both imported and domestic ones) as intermediate commodities in the production process with Leontief technology. Each activity can produce more than one commodity. In Kenya SAM, household as activity accounts (cash crop and food) are producing more than one commodity. The cost of production is calculated considering indirect tax (activity tax), which the government imposes. The final production cost (value of production) is given by the activity price, including all the activity costs considering the production level.

$$PA_a(1 - ta_a)QA_a = PVA_aQVA_a + PINTA_aQINTA_a \tag{A1}$$

$$QVA_a = iva_aQA_a \tag{A2}$$

$$QINTA_a = inta_aQA_a \tag{A3}$$

$$QINT_{c\ a} = ica_{c\ a}QINTA_a \tag{A4}$$

$$QVA_a = \propto_a^{va} \left( \sum_{f \epsilon F} \delta_{fa}^{va} QF_{f\ a}^{-\rho_{fa}^{va}} \right)^{-\frac{1}{\rho_{fa}^{va}}} \tag{A5}$$

Equation (A1) describes the cost of production calculated as the result of the profit maximization. In this equation, the right-hand side shows the separated cost of production, which is the sum of factors of production with value-added price and the aggregated intermediate input with an aggregated intermediate price. The intermediate price is the aggregation of the composite commodity price for each intermediate commodity, which will be discussed further on. The left-hand side of Equation (A1) is the value of the production defined as the activity price times the production level regarding the tax payment (activity tax) to the government account. Equation (A5), on the other hand, shows the technology of using the factors in the production in the form of constant elasticity of substitution function (CES) to produce the value-added level needed to produce the desired production level.

$$PA_a = \sum_{c \epsilon C} PXAC_{a\ c}\ \theta_{a\ c} \tag{A6}$$

$$PXAC_{a\ c} = PX_cQX_c \left( \sum_{a \epsilon A} \delta_{a\ c}^{ac} QXAC_{a\ c}^{-\rho_c^{ac}} \right)^{-1} \delta_{a\ c}^{ac} QXAC_{a\ c}^{-\rho_c^{ac}-1} \tag{A7}$$

$$QX_c = \propto_c^{ac} \left( \sum_{a \epsilon A} \delta_{a\ c}^{ac} QXAC_{a\ c}^{-\rho_c^{ac}} \right)^{-\frac{1}{\rho_c^{ac}-1}} \tag{A8}$$

It is worth mentioning that two or more different activities can produce commodities. In Kenya, water, coffee, and commodities inside the aggregated "rest of the agriculture" are produced by different activities having activity-specific prices. Equations (A6)–(A8) are the modifications in case two or more activities produce one commodity. Equation (A7) shows the technology function (CES) of the output aggregation for the *c*-th commodity produced by more than one activity.

According to Equation (A9), the factor demands of the activities in the production process are the result of the maximization problem of each activity. Due to the lack of information on factor endowment (FEND), the observed value is counted as the fixed initial factor endowment. The factor wages and the rental prices are set endogenously to guarantee the equalization of the factor supply and the factor demand from the overall activities of the economy.

$$WF_f\overline{FEND_{f\ a}} = PVA_a(1 - tva_a)QVA_a \left( \sum_{f \epsilon F} \delta_{fa}^{va}QF_{f\ a}^{-\rho_{fa}^{va}} \right)^{-1} \delta_{fa}^{va}QF_{f\ a}^{-\rho_{fa}^{va}-1} \tag{A9}$$

*Appendix A.2. Commodity Market*

Based on the Kenyan SAM, all the commodities are traded in the market, including imported ones. Looking at the relative price of import and domestically produced commodities, the producers decide the share of import and domestic commodities according to the Armington assumption. Similarly, the level of the commodities produced for the export is based on the relative prices in export and domestic sales using a CET function technology (constant elasticity of transformation). The flow of composite commodities to domestic activities is via intermediate demand of the activities and to domestic agents via final consumption demands of the agents. Consumer price includes the activity prices, the sales tax, and the tariff taxes imposed by the government and accounts for any possible marginal costs.

$$PDS_c QD_c + PE_c QE_c = PX_c QX_c \tag{A10}$$

$$QX_c = \alpha_c^t \left( \delta_c^t QE_c^{\rho_c^t} + \left(1 - \delta_c^t\right) QD_c^{\rho_c^t} \right)^{\frac{1}{\rho_c^t}} \tag{A11}$$

$$\frac{QE_c}{QD_c} = \left( \frac{PE_c}{PDS_c} \frac{1 - \delta_c^t}{\delta_c^t} \right)^{\frac{1}{\rho_c^t - 1}} \tag{A12}$$

$$PE_c = Pwe_c(1 - te_c)EXR - \sum_{c' \epsilon CT} PQ_{c'} ice_{c'\ c} \tag{A13}$$

While output commodity could be directed to domestic sale or export, the allocation of output commodity sale is obtained through the maximization of Equation (A10) subject to Equation (A11). Equation (A11) shows the assumption of imperfect transformation between domestically sold and exported commodities in the form of a CET aggregation function. For any commodity entitled for export, Equation (A12) shows the export—domestic ratio. In Kenya, water, construction, and electricity commodities have no transaction with the rest of the world. In the absence of export margins and tax d, the export price, as from Equation (A13), is used for the maximization problem. The world price of every commodity is implicitly driven by the calibration process presented in Equation (A13). This study assumes that the exchange rate (EXR) in Equation (A13) is fixed.

As it is mentioned before, the allocation of the share of import and the domestic product is modeled according to the Armington assumption by the minimization of Equation (A14) subject to Equation (A15), where, according to a CES aggregation technology function of imported commodity and domestically produced commodity, a composite commodity (QQ) of the *c*-th commodity is produced. As regards the export problem, Equation (A15) also shows the imperfect substitution between imported product and domestic commodity. Equation (A16) represents the import–domestic ratio. Based on the data, in the case of import product, the fixed ad valorem import tax rate is defined according to Equation (A17), where the import margin is the cost of transporting the imported product from the border to the domestic demand. In this case, the world price and the exchange rate are calculated the same way as for the exports.

$$PDD_c QD_c + PM_c QM_c = PQ_c(1 - tq_c)QQ_c \tag{A14}$$

$$QQ_c = \alpha_c^q \left( \delta_c^q QM_c^{-\rho_c^q} + \left(1 - \delta_c^q\right) QD_c^{-\rho_c^q} \right)^{\frac{1}{\rho_c^q}} \tag{A15}$$

$$\frac{QM_c}{QD_c} = \left( \frac{PDD_c}{PM_c} \frac{\delta_c^q}{1 - \delta_c^q} \right)^{\frac{1}{\rho_c^q + 1}} \tag{A16}$$

$$PM_c = Pwm_c(1 + tm_c)EXR + \sum_{c' \epsilon CT} PQ_{c'} icm_{c'\ c} \tag{A17}$$

In Equation (A14), PDD means the domestic demand price (for commodities produced domestically and sold domestically), and PDS is the domestic supply price in Equation (A10) (for the commodities supplied domestically). In the absence of the information on trade and transport costs within the industries, by assumption, they are

equal. Domestic demand price is the domestic supply price plus the transportation or transaction cost.

*Appendix A.3. Institutions*

The institution block in the economy of Kenya consists of one representative household, representative enterprises, a representative government, and the rest of the world account.

Households in this economy consume the home production with activity prices and marketed commodity with commodity prices, which includes the sale and tariff taxes plus the margins. Household consumption is modeled according to a linear expenditure system (LES) by solving the maximization problem of its Stone–Geary utility function [48]. The flow of income to the household is from factor payments by activities, either directly or indirectly through the enterprises or transfers from other institutions. Instead, the expenses of a household go directly to commodity consumption, direct tax payment, saving, and transfers to other institutions.

$$PQ_c QH_{c\,h} = PQ_c \gamma^m_{c\,h} + \beta^m_{c\,h}\left(EH_h - \sum_{c'\epsilon C} PQ_{c'} \gamma^m_{c'h} - \sum_{a\epsilon A}\sum_{c'\epsilon C} PXAC_{a\,c'}\gamma^h_{a\,c'h}\right) \tag{A18}$$

$$EH_h = (1 - \sum_{i\epsilon INSDNG} shii_{i\,h})(1 - MPS_h)(1 - TINS_h)YI_h \tag{A19}$$

$$PXAC_{a\,c}QHA_{a\,c\,h} = PXAC_{a\,c}\gamma^h_{a\,c\,h} + \beta^h_{a\,c\,h}(EH_h \\ - \sum_{c'\epsilon C} PQ_{c'}\gamma^m_{c'h} - \sum_{a\epsilon A}\sum_{c'\epsilon C} PXAC_{a\,c'}\gamma^h_{a\,c'h}) \tag{A20}$$

Equation (A19) shows the net income of the household considering the deduction of the fixed marginal propensity of saving ($MPS_h$), the share of net income transfer from household to other nongovernmental institutions ($shii_{i\,h}$), and the direct tax rate ($TINS_h$), from gross income of household ($YI_h$) that must be equal to the overall expenditure of the household ($EH_h$). Equation (A18) presents the first-order condition of the utility maximization of the household in the case of the consumption of marketed commodity, where $\gamma^m_{c\,h}$ is the subsistence level of consumption of marketed commodity and $\beta^m_{c\,h}$ is the marginal share of marketed commodity consumption $c$ from other commodities on the market.

One of the distinct features of Kenya SAM is the household production for its own consumption. Those commodities are not entering the market, and their value is not affected by the tariff and sales taxes. This feature has been anticipated in Equation (A20), where $QHA_{a\,c\,h}$ is the consumption quantity of the home-produced commodity, $\gamma^h_{a\,c\,h}$ is the subsistence level of home commodity consumption, and $\beta^h_{a\,c\,h}$ is the marginal share of the consumption of the *c-th* home commodity from activity *a*. In Kenya, the home commodities are produced by the household as activity food.

$$QG_c = \overline{GADJ}\overline{qg_c} \tag{A21}$$

$$YG = \sum_{i\epsilon INSDNG} TINS_i YI_i + \sum_{f\epsilon F} tf_f YF_f + \sum_{a\epsilon A} tva_a PVA_a QVA_a + \\ \sum_{a\epsilon A} ta_a PA_a QA_a + \sum_{c\epsilon CM} tm_c pwm_c QM_c EXR + \sum_{c\epsilon CE} te_c pwe_c QE_c EXR + \\ \sum_{c\epsilon C} tq_c PQ_c QQ_c + \sum_{f\epsilon F} YIF_{gov\,f} + trnsfr_{gov\,row}EXR \tag{A22}$$

$$EG = \sum_{c\epsilon C} PQ_c QG_c + \sum_{i\epsilon INSDNG} trnsfr_{i\,gov}\overline{CPI} \tag{A23}$$

Moving to the government accounts, Equation (A21) shows its consumption demand. Government demand for each *c*-th commodity in the model is fixed and is equal to an exogenous consumption quantity (qgc) times the government adjustment factor (GADJ), which is also exogenous, and it is based on the default macro closure of the model. Equation (A22) represents the income of the government, which is equal to the collected taxes plus the transfers from the other institutions. Therefore, Equation (A23) presents the final expenditure of the government, which must be equal to the transfers to the other nongovernmental domestic institutions and the marketed commodity consumption by the government itself. Thus, in this model, the government collects the tax paid fixed at an ad valorem

rate plus the transfers from the other institutions as its income. The government expenditure for the marketed commodity is formulated by fixed quantity with activity—specific prices and transfers to the other institutions by the consumer price index (CPI). The deficit of the Kenyan government is also calculated as the difference between its consumption and income.

The enterprises do not consume: their income is from receiving transfers from the other institution, and their expenses are like the household expenditure account excluding commodity consumption. Therefore, in Equation (A24), the income of any nongovernmental institutions could be seen, including households and enterprises. Similarly, Equation (A25) shows the factor income by both nongovernmental institution accounts, considering the fixed exchange rate and factor tax. The factor income from production sectors is presented by Equation (A26).

$$YI_i = \sum_{f \epsilon F} YIF_{i\,f} + \sum_{i' \epsilon INSDNG'} TRII_{i\,i'} + trnsfr_{i\,gov}\overline{CPI} + trnsfr_{i\,row}EXR \quad \text{(A24)}$$

$$YIF_{i\,f} = shift_{i\,f}\left[\left(1 - tf_f\right)YF_f - trnsfr_{row\,f}EXR\right] \quad \text{(A25)}$$

$$YF_f = \sum_{a \epsilon A} WF_f\overline{FEND_{f\,a}}QF_{f\,a} \quad \text{(A26)}$$

$$QINV_c = \overline{IADJqinv_c} \quad \text{(A27)}$$

In this model, the investment demand of every commodity is exogenous. As regards government consumption demand, the investment demand is defined as the exogenous adjustment factor multiplied by the reference year quantity of the investment for the different commodities. The exogenous investment demand allows for imbalances in the investment accounts according to the Johnson approach to market closure [49,50].

As far as macroeconomic closure goes in the CGE model developed in this study, the government consumption, the real investment of the marketed commodities, and foreign saving, which is the tradeoff between import and export of the marketed commodity, are fixed. As a numeraire in the CGE model, we adopt the consumer price index defined in Equation (A28). This is equal to the summation of the composite price of every commodity times its weight in the consumer price index:

$$\overline{CPI} = \sum_{c \epsilon C} PQ_c cwts_c \quad \text{(A28)}$$

*Appendix A.4. Market Clearing Conditions*

The market clearing conditions are the conditions imposed to have a definite equilibrium throughout the economy between all the players.

$$\sum_{a \epsilon A} QF_{f\,a} = \overline{QFS_f} \quad \text{(A29)}$$

Within the varied factors of production (labor, capital, land, and livestock), Equation (A29) implies that the fixed quantity of factor endowment must be equal to the factors used in the process of production. In this model, the quantity of the factor's endowment is set at the exogenous observed level in the SAM of Kenya. In the commodity market, the market clearing condition in the commodity market is:

$$QQ_c = \sum_{a \epsilon A} QINT_{c\,a} + \sum_{h \epsilon H} QH_{c\,h} + QG_c + QINV_c + QT_c \quad \text{(A30)}$$

According to Equation (A30), the total composite commodity produced (imported and domestic commodity) must equal the overall demand for the commodities in the economy. The right-hand side of the equation shows the total demand of the marketed commodity, which is defined as the sum of the intermediate demands, the household marketed commodity consumption, the government consumption, the investment demand, and the rest of the world demand (export demand).

In the rest of the world account, Equation (A31) depicts the market clearing condition regarding the foreign currency. The left-hand side of the equation could be seen as the value of the import plus the transfer between the factor accounts with the rest of the world that must be equal to the right-hand side of the equation. The right side of the equation could be seen as the value of export plus the value of transfers to domestic institutions and the foreign saving. The foreign saving is the tradeoff between the value of import and export, which is considered exogenous in this model with the foreign currency.

$$\sum_{c\epsilon CM} pwm_c QM_c + \sum_{f\epsilon F} trnsfr_{row\,f} = \sum_{c\epsilon CE} pwe_c QE_c + \sum_{i\epsilon INSD} trnsfr_{i\,row} + \overline{FSAV} \tag{A31}$$

Equation (A32) represents the government budget balance, where the government's income on the left side must be equal to the consumption plus the government's saving.

$$YG = EG + GSAV \tag{A32}$$

$$TINS_i = \overline{tins_i}\big(1 + \overline{TINSADJ}tins01_i\big) + \overline{DTINS}t_i \tag{A33}$$

Equation (A33) shows the direct tax balance of the domestic institutions for the government. Although the direct tax rate over the domestic institutions is fixed, the equation itself considered the situation in which the exogenous elements on the right-hand side could be set as flexible variables, such as direct tax rate $\overline{tins_i}$, direct tax scaling factor $\overline{TINSADJ}$ (which is zero in the baseline), and change in domestic institution tax rate $DTINS$ (which is also zero at the baseline). The value of $tins01$ is between 0 and 1, meaning fixed or flexible direct tax rate, respectively. In this model, we assume that institutions potentially have flexed direct tax rates; hence, $tins01$ is equal to 1.

Equation (A34) represents the balance of the saving rate of the nongovernmental institution. While Equation (A34) looks like Equation (A33) for the direct tax collection, the change in domestic institution saving rate (DMPS) can be flexible for the type of market closure applied in this model.

$$MPS_i = \overline{mps_i}\big(1 + \overline{MPSADJ}mps01_i\big) + DMPSmps01_i \tag{A34}$$

$$\sum_{i\epsilon INSDNG} MPS_i(1 - TINS_i)YI_i + GSAV + EXR\overline{FSAV} = \sum_{c\epsilon C} PQ_c QINV_c \tag{A35}$$

The investment accounts balance for the marketed commodities is represented by Equation (A35). In this equation, the domestic nongovernmental institution saving plus the government saving and the foreign saving (on the left-hand side) must equal the overall investment of the marketed commodities (on the right-hand side of the equation). As it can be noticed, on the left side of the equation, the only parameter that could make this equilibrium condition verified is the one within the balancing condition of the MPS, which is the nonexogenous parameter of DMPS in Equation (A34) (domestic institution saving rate) that is equal to zero for the based model.

Finally, the total nominal absorption is the overall commodity demands of the economy. This variable, TABS in Equation (A36), is equal to the commodity demanded by the household, including the household production for its consumption, the government consumption demand, and the investment demand of the marketed commodity. In this equation, except for private home consumption, all the variables are calculated at the marketed commodity price. The value of the TABS plus the import and the export tradeoff can be seen as the whole GDP of the economy.

$$\begin{aligned} TABS = \sum_{h\epsilon H} \sum_{c\epsilon C} PQ_c QH_{c\,h} + \sum_{a\epsilon A} \sum_{c\epsilon C} \sum_{h\epsilon H} PXAC_{a\,c} QHA_{a\,c\,h} \\ + \sum_{c\epsilon C} PQ_c QG_c + \sum_{c\epsilon C} PQ_c QINV_c \end{aligned} \tag{A36}$$

## Appendix B. Scenario A

**Table A6.** Hicks-neutral technological shock (reduction in efficiency parameter).

| Sectors | Arndt and Tarp (2001) for 20% Output Loss | This Study for 10% Output Loss |
|---|---|---|
| Basic food crops | 0.75 | 0.25 |
| Dairy | 0.85 | 0.45 |
| Rest of agriculture | 0.67 | 0.27 |

*Appendix B.1. Simulation Results of Scenario A*

Appendix B.1.1. Scenario A: The Effect of a Drought

In this section, we present the impact of drought on the Kenyan economy. As explained in Section 4, we assumed that the drought negatively affects the productivity of the agriculture sectors, leading to a reduction of 10.2% of the agricultural domestic outputs (on average).

**Table A7.** Percentage change in marketed commodity flows.

| Marketed Commodities | Drought Shock | | | | | |
|---|---|---|---|---|---|---|
| | Domestic Output | Composite Commodities | Market Price | Domestic Sales | Import | Export |
| Coffee | −18.69 | −15.90 | −0.16 | −18.67 | −15.29 | −18.69 |
| Food crops | −7.74 | −4.64 | 3.45 | −7.56 | 5.22 | −11.52 |
| Meat–dairy–livestock | −6.69 | −5.20 | 4.94 | −6.59 | 17.87 | −11.58 |
| Rest of the agriculture | −8.20 | −3.50 | 2.06 | −7.10 | 5.18 | −9.82 |
| Textile and clothing | 2.99 | −2.74 | 0.30 | −0.29 | −3.06 | 4.37 |
| Leather and footwear | −0.26 | −1.11 | −1.46 | −0.52 | −2.73 | 5.22 |
| Petroleum | 0.53 | −1.65 | 0.51 | −0.06 | −2.03 | 2.97 |
| Metals and machines | 0.15 | −0.40 | −0.98 | −0.16 | −1.50 | 4.09 |
| Chemicals | 0.33 | −3.16 | 0.38 | −0.28 | −3.79 | 4.82 |
| Fertilizers—nitrogen | −3.08 | −9.05 | −1.15 | −3.91 | −11.57 | 7.84 |
| Fertilizers—phosphorus | −5.37 | −9.57 | −1.01 | −6.26 | −11.70 | 2.24 |
| Fertilizers—potassium | −6.17 | −9.64 | −1.09 | −7.02 | −11.75 | 0.59 |
| Nonmetallic products | 2.40 | −0.14 | 0.59 | 1.40 | −0.51 | 4.21 |
| Rest of the manufacturing | 1.32 | −1.08 | −0.38 | 0.49 | −2.64 | 5.37 |
| Water | −1.80 | −1.80 | 1.99 | −1.80 | 0.00 | 0.00 |
| Electricity | −1.00 | −1.00 | −3.54 | −1.00 | 0.00 | 0.00 |
| Construction | −0.12 | −0.12 | −1.33 | −0.12 | 0.00 | 0.00 |
| Trade | 1.28 | 1.00 | −2.63 | 1.02 | −0.79 | 2.87 |
| Transport | −0.18 | −0.87 | −4.69 | −0.75 | −3.67 | 2.26 |
| Rest of services | −0.57 | −0.65 | −3.76 | −0.62 | −2.99 | 1.80 |

Since the drought directly affects agriculture, these commodities register the most substantial reduction. As shown in Table A7, food crops, dairy, and the rest of the agriculture have losses in terms of marketed commodity flow of 7.74%, 6.69%, and 8.2%, respectively. This reduction in agricultural production is coupled with an increase in their prices, generating an extra burden on both agriculture and any related sector. For instance, fertilizer production, which plays a crucial role in agricultural productivity, shrunk by 3.08% (nitrogen based), 5.37% (phosphorus based), and 6.17% (potassium based). Although the agricultural product imports increase, the reduction of the composite commodities in agriculture demonstrates the fact that the country is suffering from the lack of agrifood commodities. Hence, Table A7 shows the devastating impact of drought on agriculture commodity production and export, which are the backbone of the Kenyan economy.

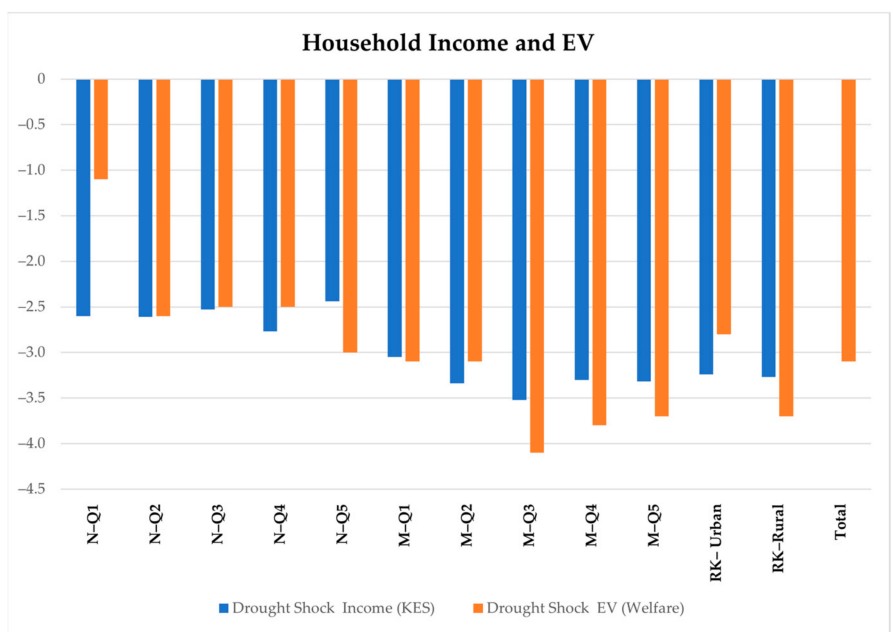

**Figure A1.** Percentage change in household's units in terms of income (in blue) and EV (welfare) (in orange). N means Nairobi, M refers to Mombasa, whereas RK describes the rest of Kenya. The households in Nairobi and Mombasa are divided in quantile, from the richest (Q1) to the poorest (Q5).

The shrink of the agricultural production and the related sectors due to the drought differently affects the Kenya population. According to Figure A1, the rural population suffers more from the drought than the urban communities in terms of both consumption and welfare (−0.9%). This result is because most of the population lives in the rural areas, where the consumption patterns highly involve agriculture products. Moreover, the drought has heterogeneous effects also on households living in the same city but having different income/wealth levels. Indeed, the middle and lower quantiles of the population in both Mombasa and Nairobi face stronger losses than the first two quantiles (richer households).

Table A8 shows the aggregate impact of drought because of climate change on the economy. The reduction in agricultural productivity negatively affects both the total absorption (−2.12%) and the private consumption (−3.05%). Moreover, the Kenyan economy registers a depletion of the trade balance with a reduction in import and export by −1.36% and −2.59%, respectively. Finally, the GDP reduces by −2.47% (reduction of 132.22 billion KES). The result of this scenario implementing the drought shock suggests that mild to severe drought could impact the economy of Kenya. This event could increase poverty and reduce domestic income flows. For countries like Kenya, where most of the population is vulnerable, especially in rural areas, the increase of poverty and the reduction of consumption may have disastrous consequences on the population livelihood.

**Table A8.** Percentage change in real GDP and national accounts, scenario A.

| Real GDP and National Accounts | Drought Shock |
|:---:|:---:|
| Absorption | −2.12 |
| Private consumption | −3.05 |
| Exports | −2.59 |
| Imports | −1.36 |
| GDP (at market prices) | −2.47 |
| Exchange rate | 0.93 |

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
