# Peer review of "Less Water, Less Oil: Policy Response for the Kenyan Future, a CGE Analysis"

_sustainability, doi:10.3390/su141811273_

Round 1
Reviewer 1 Report
This study investigates the impact of two contemporaneous covariant sudden shocks (i.e. drought and price oil shock) and the possible coping strategies through a static CGE model for Kenya. The topic of the article is interesting, and the research results are useful for guiding policy making and coping with sudden crises. We have some questions and suggestions below:
1. How does the model established in this paper verify the rationality of simulation results
2. Line 247: Kenyan Government receives a financial aid of Ksh 130bn ($ 1.16bn). What's the basis for this? How much it accounts for the government's finances.
3. This paper lacks discussion on the results and causes. The author should fully discuss the results and phenomena in this paper based on the existing relevant understanding and research, so as to prove the rationality and uncertainty of the results.
Author Response
Please, see the attachment.

Reviewer 2 Report
It was a pleasure reading the manuscript entitled “Less water, Less oil: Policy response for the Kenyan future. A 2 CGE analysis”. The topic of the article is interesting and presents a good topic for readers of this Journal
The study is clear and the methods seemed reliable to me. I have only some observations to point out.
The authors must indicate what does it article add to the subject area compared with other published material.
The units of the variables presented in the equations must be indicated.
Author Response
Please, see the attachment.

Reviewer 3 Report
I found the topic of the study very interesting and in line with the scope of the journal. To improve the overall quality of the manuscript, I have some suggestion/comments as below:
The quality of the Figure 1. Output production process in CGE model for Kenya, may be improved, at least in my pdf it is distorted a bit.
The Kenyan Social Accounting Matrix (SAM) is not clear and needs to be better explained in the text.
Section 4. Data and Simulations is very dense and requires an expert knowledge of the subject. It would be advisable to explain better.
Need a better explanation in table 3 to 8. It is hard to understand it and you should comment on the values of indicators, Percentage change in households units in terms of consumption and welfare, Scenario A (drought) and Scenario B (drought with policy response), percentage change in real GDP and national accounts, percentage change in marketed commodities flows, percentage change in import and export prices, percentage change in households units in terms of consumption and welfare and percentage change in real GDP and national accounts.
English needs to be revised.
Author Response
Please, see the attachment.

Round 2
Reviewer 3 Report
The revised version is well-written, scientifically conducted and the conclusions were comprehensively supported by the data, therefore, the revised version can be accept in present form.